# Association of Chrononutrition Indices with Anthropometric Parameters, Academic Performance, and Psychoemotional State of Adolescents: A Cross-Sectional Study

**DOI:** 10.3390/nu15214521

**Published:** 2023-10-25

**Authors:** Mikhail F. Borisenkov, Tatyana A. Tserne, Sergey V. Popov, Vasily V. Smirnov, Olga I. Dorogina, Anna A. Pecherkina, Elvira E. Symaniuk

**Affiliations:** 1Department of Molecular Immunology and Biotechnology, Institute of Physiology of Federal Research Centre “Komi Science Centre of the Urals Branch of the Russian Academy of Sciences”, 167982 Syktyvkar, Russia; cerne_tatyana@mail.ru (T.A.T.); s.v.popov@inbox.ru (S.V.P.); smirnowich@yandex.ru (V.V.S.); 2Ural Institute of Humanities, Ural Federal University, 620000 Yekaterinburg, Russia; dorogina_olga@mail.ru (O.I.D.); e.e.symaniuk@urfu.ru (E.E.S.)

**Keywords:** chronobiology disorders, feeding behavior, academic performance, depression, body weight, adolescents

## Abstract

Adolescents are an at-risk group for circadian misalignment. The contribution of sleep–wake rhythm instability to the psychoemotional, cognitive, and weight disorders of adolescents has been studied in sufficient detail. At the same time, there is insufficient information about the association between chrononutrition indices and the well-being of adolescents. The aim of this study is to investigate the relationship between chrononutrition indices and academic achievement, psychoemotional state, and anthropometric indicators in adolescents. The study involved 12,759 students in grades 6–11 of secondary schools, aged 14.2 ± 1.7 years old; 57.2% of whom were girls. Participants provided personal data, frequency and time of meals during the day and at night, on weekdays and weekends, and completed the Zung Self-Rating Depression Scale and the Yale Food Addiction Scale. There is a U-shaped association between eating mid-phase (EPFc), eating jetlag (EJL), and eating window (EW) with GPA, ZSDSI, and FA. At the same time, the frequency of night eating (NE) is linearly associated with the studied parameters. NE is the strongest predictor of ZSDSI (*β* = 0.24), FA (*β* = 0.04), and GPA (*β* = −0.22). EPFc, EJL, and EW practically do not differ in the strength of their association with the studied indicators. ZSDSI is most closely associated with the chrononutrition indices. There is a weak negative association between BMI and EW (*β* = −0.03) and NE (*β* = −0.04). Thus, circadian eating disorders are more often observed in adolescents with poor academic performance, high levels of depression, and food addiction.

## 1. Introduction

The circadian system (CS) is critical to the correct functioning of the human body and its adaptation to living in a changing environment throughout a 24 h period [1]. The role of the CS in human life did not diminish even after most people moved to cities with a favorable social environment and rooms with a comfortable microclimate. This is evidenced by numerous experimental, clinical, and population studies, which indicate that CS misalignment is accompanied by a significant deterioration in well-being, cognitive decline, and other disorders [2,3]. Prolonged exposure to CS misalignment leads to an increased risk of developing some chronic diseases and an accelerated aging process [2,4,5,6].

The most important characteristic of the CS in humans is the period of its endogenous rhythm. This characteristic is determined by genes and is an individual trait that determines a person’s chronotype. Only about 30% of people with an intermediate chronotype have a period of endogenous rhythm of approximately 24 h, and individuals with early and late chronotypes have a period of endogenous rhythm significantly less or more than 24 h, respectively [7]. External synchronizing signals, with a period of 24 h, are needed for the CS of people with a non-24 h endogenous rhythm to function normally. Light appeared to be the main external signal for the synchronization of the human CS with a 24 h rhythm [8,9].

One more important external synchronizing signal for the human CS is supposed to be food intake during the day [10]. The effect of diet on CS function has mainly been studied in experiments on animals. It was revealed that animals have a so-called «food-entrainable oscillator» in their bodies [11]. This physiological mechanism allows animals to predict the time of their next meal and thus prepare their digestive organs for food intake [11,12]. The features of this mechanism in humans have been studied recently. Eating at a later time has been found to be associated with a shift in the phase of melatonin production, which is a marker of the CS’s rhythm [13]. There are similarities between the 24 h eating pattern and the 24 h sleep–wake routine [14,15], and disturbances in these rhythms were shown to have a similar effect on the functions of the human body. Excessive shortening [16] and lengthening [17] of the eating period, skipping breakfasts, and shifting the food intake to a later time in the day [18,19,20], as well as irregular meal timing [21,22] and “eating jetlag” [15,20,23], lead to a deterioration in the quality of sleep and well-being.

The psychoemotional state and cognitive functions of healthy people are subject to cyclical changes over a 24 h period [24,25]. The mismatch between the sleep–wake rhythm phase and the central pacemaker phase is accompanied by a deterioration of the psychoemotional state in healthy people [24] and patients [26]. However, recently, data have been obtained on the important role of the rhythm of nutrition in maintaining mental health. It is known that night shift workers have an increased risk of depression [27]. In an experimental study simulating the night shift, the psychoemotional state of young people was studied in two groups; in one the waking and eating phase occurred at night, and in the other it occurred only the waking phase. As a result, it was shown that only in the first group was there a deterioration of the psychoemotional state [28], indicating that eating at night is a risk factor for depression. Breakfast skipping is associated with depression [29] and decreased academic performance [30] in adolescents.

Recent studies on chrononutrition have mainly focused on adults. There are not enough studies on the 24 h rhythm of eating in children and adolescents, who differ from adults in several physiological and psychological characteristics. Adolescents are an at-risk group for CS dysfunction. This is due to the mismatch between the sleep–wake rhythm and the rhythms of social life. People between the ages of 10 and 20 may experience a steady delay in the sleep–wake rhythm, which manifests itself in increasingly later times of both falling asleep and waking up [31]. This occurs against the backdrop of an ever-increasing academic load caused by preparation for final exams. All of this leads to instability in the 24 h sleep–wake rhythm phase during the calendar week, which is called social jetlag [32]. From 40.1% [33] to 86.4% [34] of young people are affected by social jetlag. It has been previously shown that CS mismatch in adolescents is accompanied by decreasing academic performance [35], worsening psychoemotional state [36], and an increasing risk of obesity [37]. In childhood and adolescence, the human body grows rapidly, which can affect eating behavior (a tendency to consume high-calorie foods due to the large energy needs of a growing body) as well as psychoemotional state (dissatisfaction with one’s appearance due to an imbalance of body proportions caused by different growth rates of individual organs [38]). Thus, the association between sleep–wake rhythm disturbances and the psychoemotional state, cognitive functions, and anthropometric indicators of adolescents has now been studied in sufficient detail. At the same time, there is not enough information about the relationship between the chrononutrition indices and the well-being of adolescents. It can be assumed that the association of the chrononutrition indices with anthropometric indicators and psychoemotional state has its own special features among children and adolescents.

The aim of the study is to investigate the relationship between the chrononutrition indices and academic achievement, psychoemotional state, and anthropometric indicators in adolescents.

## 2. Materials and Methods

### 2.1. Research Subject and Study Design

The study was conducted in 2019–2022 in two regions: Yekaterinburg and Komi. A total of 127 schools located in Yekaterinburg and 73 schools located in 58 settlements of the Komi Republic were randomly selected for the study. All schools worked in full-time mode during the survey period. Adolescents in grades 6 to 11 completed online tests, prepared in Google Forms, in a classroom under the control of their teacher or school psychologist. The latter informed students about the study. In addition, the initial page of the survey contained information about the goals of the study, voluntary participation, the ability to stop taking tests, and that the results obtained would be used anonymously for scientific publications only.

Inclusion criteria were as follows: 13–18-year-old male and female inhabitants of the Russian Federation who signed informed consent and received parental consent to participate in the study. Exclusion criteria were as follows: psychiatric and eating disorders.

Sample size was calculated in accordance with rules for multiple linear regression (sample size of at least 30 per variable in the model [39]) and logistic regression (sample size of at least 500 per model [40]) models.

13,706 questionnaires were obtained in total; however, 947 (6.9%) of them were excluded from further analysis because of omissions or errors. The final number of study participants was 12,759 (57.8% were female), and their average age was 14.2 ± 1.7 years.

This study was approved by the Ethics Committee of the Institute of Physiology of the Komi Science Centre of the Ural Branch of the Russian Academy of Sciences (26 March 2019). Verbal informed consent was obtained from all study participants. Additionally, schoolchildren’s parents provided written informed consent.

### 2.2. Instruments

Personal information was obtained from each study participant, who then answered diet-related questions and completed the Zung Self-Rating Depression Scale (ZSDS) [41] and the Yale Food Addiction Scale for Children (YFAS-C) [42].

#### 2.2.1. Personal Information

The participants reported their age, sex, height, body weight, waist circumference, and also indicated their academic performance and place of residence. The distribution of respondents by region of residence was as follows: Yekaterinburg (*n* = 10,194) and Komi Republic (*n* = 2565). Three groups were identified according to the season: autumn (*n* = 434), winter (*n* = 2435), and spring (*n* = 9890). The reliability of the self-assessment of the height and weight of adolescents is due to the fact that, in Russian schools, every spring and autumn, school paramedics measure the anthropometric data of a student and inform them of the measurement results.

#### 2.2.2. Academic Performance

Academic performance (GPA) was assessed using the participant’s average grade for the quarter preceding the study. Based on the Russian grading system, which includes five grades, raw scores were assigned to the low (GPA_L_: 3–3.5 scores), average (GPA_M_: 3.6–4.5 scores), or high (GPA_H_: 4.6–5 scores) categories. The data obtained correctly reproduced the known association of GPA with BMI [43], indicating the fairly high reliability of the collected material.

#### 2.2.3. Anthropometric Characteristics

The body mass index (BMI) was calculated as weight in kilograms divided by height in meters squared. BMI growth charts were used to calculate BMI percentiles (BMI%), which were adjusted to sex and age [44]. Following the criteria of the World Health Organization [44], the BMI categories (BMIc) were determined: (1) underweight (*n* = 818); (2) normal weight (*n* = 10,126); (3) overweight (*n* = 1281); and (4) obese (*n* = 534). The waist-to-height ratio (WHtR) was determined as waist circumference (cm)/height (cm) [45].

#### 2.2.4. Meal Timing

Each participant answered questions regarding diet on weekdays and weekends: (1) the number of eating episodes per day; (2) the timing of each meal; and (3) the frequency of eating at night (during night awakening). Answer options (scores): never (0), 1–3 times a year (1), 1–3 times a month (2), 1–3 times a week (3), every night (4).

Based on these data, indicators characterizing meal timing were calculated using Formulas (1)–(5), which are similar to those that were proposed for assessing indicators characterizing the circadian sleep–wake rhythm and its variability during the calendar week [32,37,46]:EW_W/F_ = EN_W/F_ − E1_W/F_(1)
MEW = ((EW_W_ · WD) + EW_F_ · (7 − WD))/7(2)
EP_W/F_ = E1_W/F_ + (EW_W/F_ · 0.5)(3)
EPFc = EP_F_ − 0.5 · (EW_F_ − ME)(4)
EJL = EP_F_ − EP_W_(5)
where EW_W/F_—eating window on weekdays/weekends; E1_W/F_/EN_W/F_—time of the first/last meal on weekdays/weekends; MEW—average weekly eating window; WD—number of weekdays; EP_W/F_—eating mid-phase on weekdays/weekends; EPFc—eating mid-phase on weekends, corrected for weekdays/weekend variability; EJL—eating jetlag.

#### 2.2.5. ZSDS

The ZSDS [41], adapted for children, consisted of 20 statements describing the symptoms of depression. The ZSDS indices (ZSDSIs), varying from 25 to 100, were obtained from the sum of raw ZSDS scores, which ranged from 20 to 80 [47,48]. The following levels of depression were assigned: I, no depression (ZSDSI ≤ 50); II, minimal to mild depression (ZSDSI 51–59); III, moderate to significant depression (ZSDSI 60–69); and IV, severe to extreme depression (ZSDSI ≥ 70). Cronbach’s *α* was equal to 0.865.

#### 2.2.6. YFAS-C

The incidence of FA was evaluated using the children’s version of the YFAS (YFAS-C) [42], which consisted of 25 questions and revealed seven diagnostic criteria for substance dependence and significant impairment related to eating behavior [49]. The YFAS-C provided the symptom count (SC, range, 0–7) and a dichotomous measure of FA; respondents with three or more symptoms and significant clinical impairment or distress were diagnosed with FA. Cronbach’s *α* was equal to 0.889.

#### 2.2.7. The Validity of the Instruments Used

In our study, the ZSDS and YFAS-C scales translated into Russian were used. Comprehensive validation of Russian translations of the tests was not carried out. However, in previous studies, we have shown that (1) the Cronbach’s alpha tests have satisfactory internal consistency (this study as well as [43,50,51,52,53,54,55]); (2) there was a significant relationship between the number of symptoms of food addiction assessed using YFAS-C and the emotional type of eating behavior assessed using DEBQ [50,51], as well as a positive relationship between food addiction and the level of depression assessed using ZSDS [43,50,51,56]; and (3) the results of testing depression and food addiction using the ZSDS and YFAS positively correlate with anthropometric indicators (BMI and WHtR) [50,51,53,55,56]. In general, all this indicates the fairly high reliability of these tools.

### 2.3. Data and Statistical Analyses

The SPSS ver. 20 (SPSS, Inc., Chicago, IL, USA) package was used. The descriptive statistics of the variables, including mean, standard deviation, skewness, and kurtosis, were calculated. The distribution of BMI and WHtR differed from normal. Therefore, transformed BMI% and WHtRc indicators with a normal distribution were used in further analyses (Table 1). Student’s test was used to determine significant differences between groups of continuous variables, and the Chi-squared test was used between categorical variables. The effect size for categorical variables was measured using Cramer’s V (*φ*) [57].

Preliminary analysis showed that there was a nonlinear U-shaped or inverted U-shaped association of chrononutrition indices (EPFc, EJL, MEW, but not NE) with academic performance (GPA), psychoemotional state (ZSDSI), and food addiction (SC) (Figure 1, Appendix A). To conduct multiple regression analyses of the relationship between these indicators, an additional transformation of EPFc, EJL, and MEW was carried out using the following formula:(6)$$Y1=\surd (1+|Xn–\bar{X|})$$
where *Y*1—EPFc1, EJL1, MEW1; $\bar{X}$—average value of the indicator (EPFc, EJL and MEW); *X*n—initial value of the indicator (EPFc, EJL and MEW).

#### 2.3.1. Multiple Regression

Analyses were performed using BMI%, WHtRc, GPA, ZSDSI, and SC as dependent variables and meal-timing indices (EPFc, EPFc1, EJL, EJL1, MEW, MEW1, NE) as independent variables (predictors) adjusted for age, sex (codes: 0—female; 1—male), season (code: 1—IX-XI Months, 2—XII-II Months, 3—III-V Months), and latitude (codes: 1—South; 2—North). A stepwise inclusion procedure was used to determine the final set of predictors used in the model. The results were adjusted using Bonferroni correction for multiple comparisons. The variance inflation factor (*VIF*) was used to evaluate multicollinearity in the model [59]. A predictor was excluded if *VIF* was ≥5.

#### 2.3.2. Logistic Regression

Analyses were performed using Ov/Ob (code: 0—no Ov/Ob; 1—Ov/Ob), GPA_L_ (code: 0—GPA_M/H_, 1—GPA_L_), depression (code: 0—no-to-mild, 1—moderate-to-extreme), and FA (code: 0—no FA, 1—FA) as dependent variables, while using age, sex (code: 0—female; 1—male), season (code: 1—IX-XI Months, 2—XII-II Months, 3—III-V Months), latitude (code: 0—south, 1—north), and meal timing indices (MEW, MEW1, EJL, EJL1, EPFc, EPFc1, NE) as independent variables (predictors); a stepwise inclusion of predictors in the model was performed, and only significant factors were included in the final model. Odds ratio and 95% confidence intervals were calculated for each model. The results were adjusted using Bonferroni correction for multiple comparisons. Models’ goodness of fit was tested using the Omnibus and Hosmer–Lemeshow tests.

## 3. Results

In total, 10% and 4.2% of the examined schoolchildren were overweight and obese, whereas 6.4% were underweight. A further 2.2% of the children and adolescents showed signs of central adiposity (WHtR > 0.5; Table 2), 20.2% of schoolchildren had moderate-to-extreme depression, and 5.3% had food addiction (Table 1). MEW for the examined schoolchildren amounted to *M* (SD) 9.7 (3.9) h, EJL—1.2 (1.8) h, EPFc—15:30 (02:24) h, and NE—1.0 (1.3), which was equal to 1–3 times per year (Table 1).

There was a nonlinear U-shaped or inverted U-shaped association of the chrononutrition indices (EPFc, EJL, MEW, but not NE) with GPA, ZSDSI, and SC (Figure 1; Appendix A). Groups of schoolchildren with extremely high and extremely low values of meal-timing indicators demonstrated low academic performance, high levels of depression, and food addiction. The only exception was the NE indicator, which had a linear relationship with all studied indicators (Figure 1; Appendix A).

There was a nonlinear U-shaped association of BMI with depression (ZSDSI) and food addiction (SC) (Figure 2). Underweight and obese adolescents (categories 1 and 4, Figure 2) had higher rates of food addiction and depression. There was a linear association between GPA and BMI: lower GPA values were observed in overweight and obese schoolchildren.

The nature of the relationship between the Indicators presented in Figure 1, Appendix A, allows us to distinguish three categories of chrononutrition indices: extremely low, optimal, and extremely high (Table 3). At the boundaries of these zones, there is a significant change in almost all studied indicators.

Multiple regression (Table 4) and logistic regression analyses (Table 5) revealed a significant linear association of the anthropometric indicators with two chrononutrition indices, MEW and NE. Higher BMI values (Model 1, Table 4) and a higher frequency of ov/ob detection (Model 1, Table 5) were observed in schoolchildren with low MEW values. Children and adolescents with minimal NE values had higher BMI and WHtR values (Models 2 and 3, Table 4). Judging by the value of the ∆*R*^2^ indicator (Models 1–3, Table 4), MEW and NE are weak predictors of BMI and WHtR.

There was a significant association between GPA and all four chrononutrition indices (Models 4–7, Table 4). Schoolchildren with average values of EJL, MEW, and EPFc demonstrated the highest academic performance. The deviation of indicators from average values (decrease and increase) was accompanied by a decrease in GPA (Models 4–6, Table 4). There was a negative linear association between GPA and NE: schoolchildren with minimal NE values had the highest GPA scores (Model 7, Table 4). Logistic regression analysis revealed a significant association between GPA_L_ and MEW: schoolchildren with extreme values in the MEW indicator were most often found to have poor academic performance (Model 2, Table 5).

There was a significant association between levels of depression and food addiction and all four chrononutrition indices (Table 4 and Table 5). The maximum values of depression and food addiction were observed in schoolchildren whose EJL, MEW, and EPFc indicators deviated most strongly from the average values (Models 8–10, 12–14, Table 4, Models 3–5, 7–9, Table 5). There was a linear association between levels of depression and food addiction and the NE indicator (Models 11 and 15, Table 4, Models 6 and 10, Table 5). Judging by the value of the ∆*R*^2^ indicator, NE is the strongest predictor of GPA, ZSDSI, and SC (Models 7,11,15, Table 4), and ZSDSI is most closely associated with all four chrononutrition indices (Models 8–11, Table 4).

## 4. Discussion

A nonlinear association of the chrononutrition indices (EPFc, MEW, and EJL) with indicators of academic performance, psychoemotional state, and food addiction in children and adolescents was found in the present study. At the same time, the NE indicator was linearly related to all of the indices mentioned above. Multiple regression analysis showed that NE was the strongest predictor of academic performance, psychoemotional state, and food addiction, and that the level of depression was most closely related to all chrononutrition indices presented in this study. We also observed a weak inverse linear relationship between BMI and only two of the chrononutrition indices, MEW and NE. These findings are consistent with others that have shown that an excessively shortened eating window [16] is associated with an increased risk of obesity in adults. Irregular eating [21] and night eating [60] have previously been shown to be positively associated with depression in adults, which is also consistent with our findings.

At the same time, the study in [60] noted a positive association between nighttime eating and the risk of obesity in adults. This is not consistent with our results. Apparently, the differences in the obtained data are due to the physiological characteristics of children and adolescents. The inverse relationship we observed between NE and BMI meant that underweight children and adolescents were more likely to eat after waking up at night. Although the cross-sectional design of the study does not allow us to judge the cause-and-effect relationships between the studied indicators, it should be noted that the human body experiences intense growth in adolescence, and the need for energy and macro- and micronutrients in adolescents is much higher than in adults. Therefore, it can be assumed that underweight children and adolescents wake up more often at night from hunger since they cannot provide the body with the necessary amount of food during the day. It seems that this fact also explains the inverse relationship between BMI and MEW, which indicates that underweight children and adolescents have an increased eating window. It can be assumed that underweight adolescents increase the frequency of meals during the day and, accordingly, the MEW, to meet the needs of a growing body in terms of energy and nutrients. Overweight and obese adolescents may eat less frequently during the day and avoid eating at night since their fat reserves replenish the energy necessary for body growth.

The nonlinear U-shaped or inverted U-shaped association of the chrononutrition indices with academic performance and the psychoemotional state of schoolchildren is similar to the association of GPA [61] and depression [62] with sleep duration. It should be emphasized that the association between BMI and sleep duration is also U-shaped [61], whereas, according to our study, the risk of obesity in children and adolescents is linearly associated with the MEW. According to somnologists [61,62,63], there is an optimal sleep duration at which academic performance, psychoemotional state, and BMI are within normal limits. Insufficient and excessive sleep duration are associated with the deterioration of human cognitive functions, psychoemotional state, and anthropometric indicators.

Just as somnology determines the boundaries of the optimal sleep duration for various age groups, the science of chrononutrition should also determine the boundaries of the optimal chrononutrition indices for various age groups. We tried to identify such boundaries for children and adolescents by using academic performance, psychoemotional state, and frequency of detection of food addiction as criteria for assessing their well-being. Preliminary analysis established that the optimal MEW values range from 7.5 to 12.5 h, EJL from −1 to +3 h, and EPFc from 13.5 to 17.33 h. In further research, the boundaries of the optimal zones of these indicators for different age groups should be more strictly assessed, since it can be assumed that the nature of the relationship between these indicators and the state of the body changes significantly with age. For example, the upper limit of the optimal eating window zone, beyond which the risk of developing obesity in adults increases, was shown to be 12 h [17]. However, our data show that this rule does not apply to children and adolescents. Children and adolescents with high MEW values are classified as underweight.

The positive association of NE with FA and the ZSDSI indicates that adolescents with food addiction and depression are more likely to eat at night. These results are consistent with previous findings showing that adolescents with FA fall asleep at later times than their peers without food addiction [56]. At the same time, the positive association of NE with FA and the ZSDSI, combined with the negative association of NE and BMI, indirectly suggests that an increased risk of developing depression and food addiction may be observed in underweight children and adolescents. Our comparative analysis of the association of psychoemotional state and the frequency of detection of food addiction with the anthropometric indicators of schoolchildren partially confirms this conclusion: children and adolescents, both overweight and underweight, have an increased incidence of depression and food addiction. It was previously shown that food addiction was observed not only in overweight people [50,64], but also in people with normal weight [65]. Some studies [66,67] noted an increased incidence of food addiction not only in obese individuals but also in underweight ones. It was suggested [65] that signs of food addiction in individuals with normal body weight may be considered a risk factor for the future development of obesity. In further studies, special attention should be paid to the age-related dynamics of anthropometric indicators in people who had a combination of insufficient body weight with high levels of depression and food addiction during childhood and adolescence. It can be proposed that these people may have an increased risk of developing obesity in adulthood. At least one recent publication showed that people who progress from normal body weight in adolescence to obesity in adulthood were predisposed to an increased risk of developing psychoemotional state disorders but not eating disorders [68].

In this study, there was a slightly higher rate of detection of food addiction (5.3 vs. 4.5%) and an approximately two-times higher incidence rate of depression (20.2 vs. 12.5%) in children and adolescents than in our previous study [50]. This appears to be due to the fact that the present study was conducted during the COVID-19 pandemic. It was shown that, during the pandemic, there was an increase in the incidence of food addiction [69] and depression [70].

This study has a number of advantages and limitations. The advantage of this work is that the sample size is large enough for us to be confident in the reliability of the results of the analysis of the relationship between the studied indicators. Even though chrononutrition has been actively developing, this work is one of the few studies devoted to a systematic analysis of the association of chrononutrition indices with academic performance, psychoemotional state, and food addiction in children and adolescents. A limitation of this study is the fact that the anthropometric indicators and academic performance of the study participants were self-assessed, which reduces the reliability of the obtained data. We did not evaluate the contribution of the level of physical activity to the studied indicators, which is a significant drawback of the study design. The cross-sectional design of the study does not allow us to judge the cause-and-effect relationships between the studied indicators.

## 5. Conclusions

A nonlinear association is noted between EPFc, EJL, and EW and academic performance, psychoemotional state, and food addiction in children and adolescents. At the same time, NE is linearly associated with the studied indicators. NE is the strongest predictor of depression, food addiction, and academic performance. EPFc, EJL, and EW practically do not differ in the strength of their association with the studied indicators. The level of depression is most closely related to the chrononutrition indices. There is a weak negative association between BMI and EW and NE. Preliminary analysis establishes that the optimal MEW values range from 7.5 to 12.5 h, EJL from −1 to +3 h, and EPFc from 13.5 to 17.33 h. Thus, disturbances in the daily rhythm of eating are more often observed in adolescents with low academic performance, high levels of depression, and food addiction.

## 6. Future Studies and Practical Recommendations

### 6.1. Future Studies

In the future, it is necessary to conduct a more detailed and thorough analysis of the sex- and age-related dynamics of the studied indicators in order to identify the limits of the norm of the chrononutrition indices for different sex and age groups. It is necessary to analyze the influence of a number of external (latitude of residence, season of the year, social status, nutrition, etc.) and internal (level of physical activity, weight status, etc.) factors on the chrononutrition indices for adolescents. It is also necessary to analyze in detail the relationship between adolescents’ chrononutrition indices and indicators characterizing the sleep–wake rhythm, as well as their possible joint impact on cognitive, psychoemotional, and anthropometric indicators.

### 6.2. Practical Recommendations

1. Our previous study [71] showed that adolescents studying at afternoon school shift are at risk of developing obesity. We hypothesized that the reason for this increased risk of obesity is their increased eating window. The results presented in this paper make it possible to comprehensively study this issue and, on their basis, recommend a scientifically based approach to organizing the educational process, taking into account the peculiarities of the chrononutrition of adolescents of various age groups, and, ideally, completely abandoning classes during the afternoon shift. 2. Currently, the scientific community is actively discussing the use of the time-restricted eating (TRE) procedure as a means of preventing obesity in humans (see, for example [72]). The data presented in our study indicate that the use of TRE for adolescents has certain limitations: the excessive reduction of the eating window may cause negative consequences for psychoemotional state and cognitive function.

## Figures and Tables

**Figure 1 nutrients-15-04521-f001:**
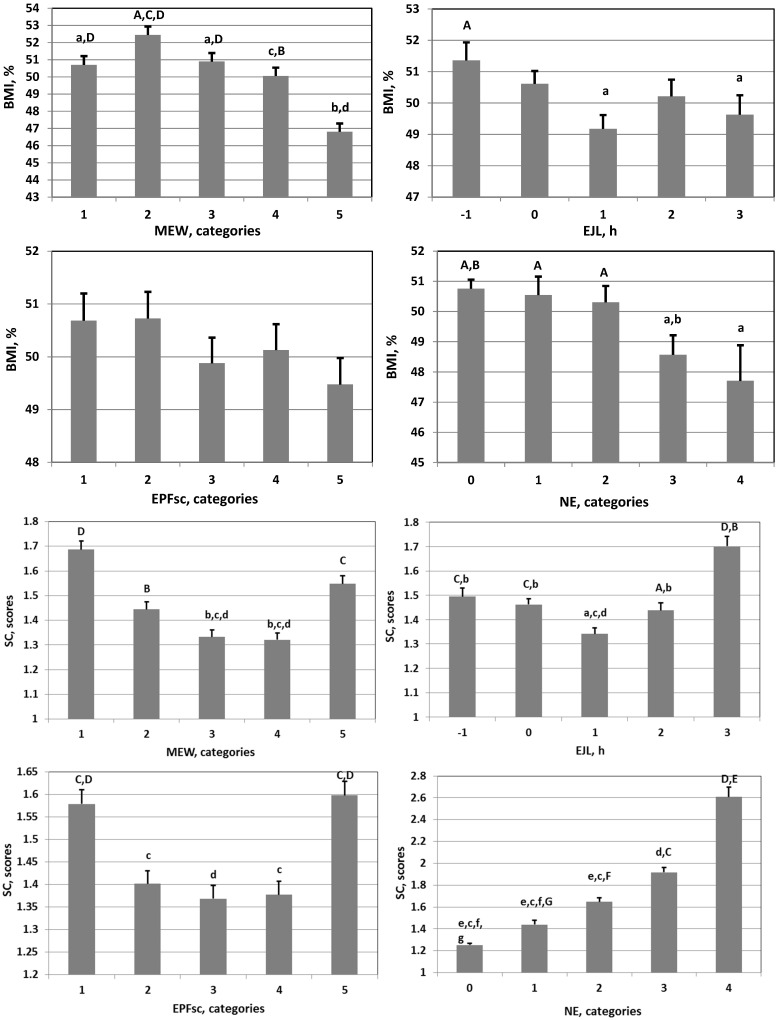
The association of meal-timing indices with BMI and SC in schoolchildren. BMI: body mass index percentiles; SC: symptom counts of food addiction; MEW: mean weekly eating window; EJL: eating jetlag; EPFsc: eating mid-phase on weekends corrected for weekday/weekend variability; NE: night eating. MEW and EPFsc were divided into five categories equal to pentiles, as described in Altman, Bland [58]. Differences between groups indicated by letters are significant (Student test: A > a, *p* < 0.05; B > b, *p* < 0.01; C > c, *p* < 0.001; D > d, F > f, G > g, *p* < 0.0001; E > e, *p* < 0.00001).

**Figure 2 nutrients-15-04521-f002:**
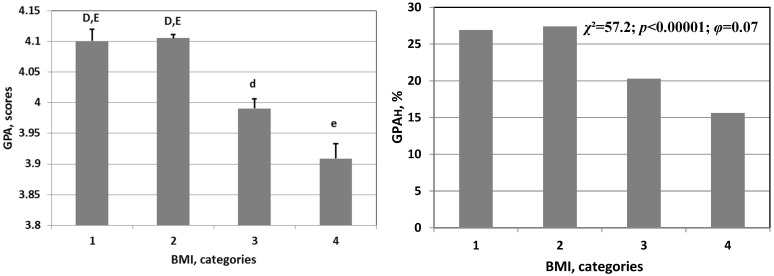
The association of continuous (left panel) and categorical indices of academic performance, food addiction, and depression with anthropometric characteristics of schoolchildren. GPA: academic performance scores; GPA_H_: incidence rate of students with a high GPA; SC: symptom counts of food addiction; ZSDSI: level of depression, scores; BMI categories: 1—underweight, 2—normal weight, 3—overweight, 4—obese. Differences between groups indicated by letters are significant (Student test: A > a, *p* < 0.05; D > d, *p* < 0.0001; E > e, *p* < 0.00001). *χ*^2^: Chi-squared test; *φ*: Cramer’s V test of effect size.

**Table 1 nutrients-15-04521-t001:** Descriptive statistics of quantitative variables.

Parameter	Abbreviation	*M*	*SD*	*S*	*K*
Age		14.20	1.70	0.10	0.20
Body mass index	BMI	19.96	4.14	8.06	170.72
	BMI%	50.25	24.09	−0.02	−0.44
Waist-to-height ratio	WHtR	0.39	0.05	0.44	2.07
	WHtRc	0.37	0.05	0.12	−0.11
Academic performance	GPA	4.09	0.53	−0.08	−0.83
Depression	ZSDSI	48.42	12.20	0.41	−0.19
Symptom counts of food addiction	SC	1.48	1.49	1.33	1.68
Night eating	NE	0.98	1.26	0.91	−0.50
Eating window	MEW	9.68	3.90	−1.11	0.38
	MEW1	1.90	0.59	0.71	−0.32
Eating jetlag	EJL	1.21	1.83	0.25	4.13
	EJL1	1.60	1.51	0.56	−0.98
Eating phase	EPFc	15.52	2.40	−0.31	1.15
	EPFc1	1.61	0.44	0.85	0.32

BMI%: body mass index, percentiles; WHtRc: the values of the indicator were divided into four categories: ≤0.29, 0.3–0.39, 0.4–0.49, and ≥0.5; MEW1, EJL1, and EPFc1 were calculated using Formula (6), see the text for explanation.

**Table 2 nutrients-15-04521-t002:** Descriptive statistics of qualitative indicators.

Parameter	Gradation	Abbreviation	N	%
All			12,759	
Sex	Females	F	7292	57.2
	Males	M	5467	42.8
BMIc	Underweight	**U**	**818**	**6.4**
	Normal weight	N	10,126	79.4
	Overweight	Ov	1281	10.0
	Obese	Ob	534	4.2
		**Ov/Ob**	**1815**	**14.2**
WHtRc		≤0.29	411	3.2
		0.3–0.39	6904	54.1
		0.4–0.49	5169	40.5
		≥0.5	275	2.2
WHtRc1	No central adiposity	WHtR < 0.5	12,484	97.8
	Central adiposity	**WHtR ≥ 0.5**	**275**	**2.2**
Academic performance	Low	**GPA_L_**	**1775**	**13.9**
	Mean	GPA_M_	7634	59.8
	High	GPA_H_	3350	26.3
Depression	No	ZSDSIc1	7200	56.4
	Minimal/Mild	ZSDSIc2	2987	23.4
	Moderate/Significant	ZSDSIc3	1977	15.5
	Severe/Extreme	ZSDSIc4	595	4.7
		**Depression**	**2573**	**20.2**
Food addiction	Yes	FA0	12,082	94.7
	No	**FA1**	**677**	**5.3**

BMIc: body mass index categorical; WHtRc: weight-to-height ratio categorical. Bold values were used as dependent variables in logistic regression analyses.

**Table 3 nutrients-15-04521-t003:** Chrononutrition indices categories.

Parameter	Extremely Low	Optimal	Extremely High
MEW, h	≤7.5	7.51–12.5	>12.5
EJL, h	≤−1	−0.99–3	>3
EPFc, h	≤13.5	13.51–17.33	>17.33
NE	–	≤1–3 per year	≥1–3 per month

MEW: mean weekly eating window; EJL: eating jetlag; EPFc: eating mid-phase on weekends corrected for weekday/weekend variability; NE: night eating.

**Table 4 nutrients-15-04521-t004:** The results of multiple regression analyses.

#	Dependent Variable	Predictor	*B*	*β*	*P*	*R* ^2^	∆*R*^2^	*VIF*
1	BMI%	MEW	−0.191	−0.031	0.001	0.001	0.001	1.005
		Sex	5.217	0.107	0.000	0.012	0.011	1.003
		Age	−1.062	−0.072	0.000	0.017	0.005	1.056
2	BMI%	NE	−0.691	−0.036	0.000	0.001	0.001	1.005
		Age	−0.968	−0.065	0.000	0.005	0.004	1.008
3	WHtR	NE	−0.002	−0.038	0.000	0.001	0.001	1.006
		Age	−0.002	−0.052	0.000	0.003	0.002	1.025
		Lat	0.005	0.035	0.001	0.004	0.001	1.047
4	GPA	EJL1	−0.130	−0.088	0.000	0.008	0.008	1.001
		Season	−0.005	−0.031	0.001	0.009	0.001	1.125
5	GPA	MEW1	−0.086	−0.095	0.000	0.009	0.009	1.005
		Season	−0.006	−0.034	0.000	0.011	0.002	1.126
6	GPA	EPFc1	−0.110	−0.091	0.000	0.008	0.008	1.007
		Season	−0.007	−0.038	0.000	0.011	0.003	1.086
7	GPA	NE	−0.086	−0.205	0.000	0.044	0.044	1.004
		Season	−0.005	−0.030	0.001	0.046	0.002	1.129
8	ZSDSI	EJL1	0.840	0.103	0.000	0.011	0.011	1.009
		Season	0.274	0.068	0.000	0.015	0.004	1.017
		Lat	−1.269	−0.042	0.000	0.017	0.002	1.029
9	ZSDSI	MEW1	2.734	0.131	0.000	0.018	0.018	1.007
		Season	0.269	0.067	0.000	0.022	0.004	1.017
		Lat	−1.340	−0.044	0.000	0.023	0.001	1.030
10	ZSDSI	EPFc1	3.449	0.124	0.000	0.016	0.016	1.010
		Season	0.267	0.066	0.000	0.020	0.004	1.018
		Lat	−1.405	−0.046	0.000	0.021	0.001	1.032
11	ZSDSI	NE	2.319	0.237	0.000	0.053	0.053	1.005
		Sex	−0.270	−0.068	0.000	0.095	0.042	1.018
		Lat	−1.616	−0.053	0.000	0.099	0.004	1.032
12	SC	EJL1	0.358	0.087	0.000	0.008	0.008	1.003
		Sex	−0.224	−0.075	0.000	0.013	0.006	1.000
13	SC	MEW1	0.253	0.100	0.000	0.010	0.010	1.006
		Sex	−0.229	−0.076	0.000	0.016	0.006	1.001
14	SC	EPFc1	0.299	0.089	0.000	0.007	0.007	1.009
		Sex	−0.230	−0.077	0.000	0.013	0.006	1.001
15	SC	NE	0.031	0.037	0.000	0.046	0.046	1.023
		Season	0.015	0.030	0.001	0.047	0.001	1.018

#: model number; BMI%: body mass index percentiles; WHtR: central adiposity index scores; GPA: academic performance; ZSDSI: level of depression scores; SC—symptom counts of food addiction; Lat—latitude of residence; EPFc—eating mid-phase on weekends corrected for weekday/weekend variability; EPFc1: transformed EPFc as described in Methods; EJL—eating jetlag; EJL1—transformed EJL as described in Methods; MEW—eating window, MEW1—transformed MEW as described in Methods; NE—night eating; a series of multiple regression were performed in which BMI% (Model 1 and 2), WHtR (Model 3), GPA (Models 4–7), ZSDSI (Models 8–11), and SC (Models 12–15) were specified as the dependent variables, while age, sex (code: 0—female; 1—male), season (code: 1—IX-XI Months, 2—XII-II Months, 3—III-V Months), and Lat (code: 0—south, 1—north), MEW, MEW1, EJL1, EPFc1, and NE were specified as independent variables (predictors); only significant variables were included in the final model. *B*: non-standardized regression coefficient; *β*: standardized regression coefficient; *P*: Bonferroni-corrected significance of *B*; *R*^2^: total variance accounted for predictors at their stepwise inclusion in the model; Δ*R*^2^: portion of the variance accounted for by separate predictors in the model; *VIF*: variation inflation factor.

**Table 5 nutrients-15-04521-t005:** The results of logistic regression analyses.

#	Dependent Variable	Predictor	*B*	ExpB	[95% CI]	^&^ *P*	Omnibus Test		Hosmer-Lemeshov Test	
							*χ* ^2^	*P*	*χ* ^2^	*P*
1	Ov/Ob	MEW	−0.030	0.971	[0.958–0.984]	0.000	326.838	0.000	10.482	0.233
		Age	−0.130	0.878	[0.849–0.908]	0.000				
		Sex	0.783	2.188	[1.966–2.435]	0.000				
2	GPA_L_	MEW1	0.148	0.878	[1.117–1.204]	0.000	407.995	0.000	15.954	0.430
		Sex	−0.878	2.188	[0.373–0.462]	0.000				
3	Depression	EJL1	0.725	2.065	[1.830–2.330]	0.000	404.923	0.000	4.184	0.840
		Age	0.042	1.043	[1.013–1.074]	0.005				
		Sex	−0.746	0.474	[0.429–0.524]	0.000				
		Season	0.027	0.974	[0.958–0.990]	0.002				
		Lat	−0.208	0.813	[0.715–0.924]	0.002				
4	Depression	MEW1	0.441	1.554	[1.440–1.678]	0.000	396.172	0.000	9.137	0.331
		Age	0.043	1.044	[1.014–1.075]	0.004				
		Sex	−0.755	0.470	[0.425–0.520]	0.000				
		Season	0.026	0.975	[0.959–0.991]	0.002				
		Lat	−0.227	0.797	[0.701–0.906]	0.001				
5	Depression	EPFc1	0.528	1.696	[1.531–1.879]	0.000	370.568	0.000	6.905	0.547
		Age	0.044	1.045	[1.015–1.076]	0.003				
		Sex	−0.758	0.469	[0.424–0.518]	0.000				
		Season	0.025	0.976	[0.960–0.992]	0.004				
		Lat	−0.235	0.790	[0.695–0.898]	0.000				
6	Depression	NE	0.316	1.371	[1.324–1.420]	0.000	581.241	0.000	4.047	0.853
		Age	0.052	1.053	[1.023–1.085]	0.000				
		Sex	−0.775	0.461	[0.417–0.509]	0.000				
		Season	0.030	0.970	[0.954–0.986]	0.000				
		Lat	−0.277	0.758	[0.666–0.862]	0.000				
7	FA	EJL1	0.677	1.968	[1.612–2.402]	0.000	122.042	0.000	5.705	0.680
		Age	0.067	1.070	[1.016–1.125]	0.010				
		Sex	−0.661	0.516	[0.430–0.621]	0.000				
		Season	0.051	0.950	[0.922–0.979]	0.001				
8	FA	MEW1	0.380	1.463	[1.279–1.673]	0.000	111.488	0.000	13.058	0.110
		Age	0.069	1.071	[1.019–1.127]	0.008				
		Sex	−0.671	0.511	[0.425–0.615]	0.000				
		Season	0.050	0.951	[0.923–0.980]	0.001				
9	FA	EPFc1	0.520	1.682	[1.407–2.012]	0.000	112.776	0.000	6.930	0.544
		Age	0.069	1.071	[1.018–1.127]	0.008				
		Sex	−0.676	0.508	[0.423–0.611]	0.000				
		Season	0.049	0.952	[0.924–0.981]	0.001				
10	FA	NE	0.307	1.360	[1.282–1.443]	0.000	178.961	0.000	11.599	0.170
		Age	0.080	1.083	[1.031–1.139]	0.002				
		Sex	−0.646	0.524	[0.438–0.627]	0.000				
		Season	0.055	0.947	[0.919–0.975]	0.000				

#—model number; Ov/Ob: incidence rate of overweight + obesity, %; GPA_L_: incidence rate of low academic performance, % (for detail see Methods); Depression: incidence rate of moderate-to-extreme depression, %; FA: food addiction, %; MEW/MEW1: eating window/transformed MEW; EJL1: transformed eating jetlag; EPFc1: transformed eating mid-phase; NE: night eating; Lat: latitude of residence; a series of logistic regression analyses were performed in which Ov/Ob (code: 0—no Ov/Ob; 1—Ov/Ob) (Model 1), GPA_L_ (code: 0—GPA_M/H_, 1—GPA_L_) (Model 2), Depression (code: 0—no-to-mild, 1—moderate-to-extreme) (Models 3–6), and FA (code: 0—no FA, 1—FA) (Models 7–10) were specified as the dependent variables, while age, sex (code: 0—female; 1—male), season (code: 1—IX-XI Months, 2—XII-II Months, 3—III-V Months), Lat (code: 0—south, 1—north), MEW, MEW1, EJL1, EPFc1, and NE were specified as independent variables (predictors). *B*: regression coefficient; OR: odds ratio; CI: confidence interval; ^&^*P*: Bonferroni-corrected significance of the regression coefficient; models’ goodness of fit was tested using Omnibus and Hosmer–Lemeshow tests.

## Data Availability

The data for this study are available on request from the corresponding author.

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
