# Peer review of "Association of Chrononutrition Indices with Anthropometric Parameters, Academic Performance, and Psychoemotional State of Adolescents: A Cross-Sectional Study"

_nutrients, 2023, doi:10.3390/nu15214521_

Round 1

Reviewer 1 Report

General Comment

An original article was made. The objective was  to study the association of chrononutrition characteristics with anthropometric indicators (BMI, WHtR), academic performance (GPA), depression (ZSDSI), and food addiction (FA) in schoolchildren. The study investigated a novel, original, relevant and scarcely studied topic. Comments and suggestions to strengthen the manuscript are presented below.

Title: The title should indicate the type of study (review strobe guidelines for cross-sectional studies)

Abstract: The justification of the study must be indicated in the abstract.

keywords: Authors should review keywords. Thesauros should be privileged. It is recommended to review the MeSH Database (https://www.ncbi.nlm.nih.gov/mesh/) 

Introduction: It is suggested to review in a little more depth the current state of research on the association between chrononutrition, school performance and emotional state of schoolchildren. Besides, the abstract and in the introduction have a different main objective. It is suggested that both be homogenized.

Methods:

·        the study design should be indicated.

·        It should be indicated how the sample calculation was done. It should be indicated what type of sampling was done. In addition, it should be mentioned how the sample was recruited.

·        On the other hand, the inclusion and exclusion criteria of the sample are not presented, they must be added.

  • The authors do not present a section that accounts for when the evaluations were made, how long the measurements took, who made them, under what conditions the measurements were made, as well as at what time of day the evaluations were made. This last information is essential for the measurement of body composition

Results: The authors provide a concise and precise description of the results of study. Congratulation.

·        just some details: Check the order of the tables. The results table cannot be in the discussion.

Discussion:

Authors discussed the results and how they can be interpreted in perspective of previous studies. Although, little is said about the findings and their implications. There is no section that accounts for the contributions and practical implications of this study. I suggest delving into the implications and practical contributions of the results.

Reviewer 2 Report

The article presents a comprehensive and well-executed study that sheds light on the critical relationship between chrononutrition characteristics and various aspects of adolescents' well-being and academic performance. The research, involving a substantial sample size of 12759 students, provides valuable insights into this important area of study.

One of the strengths of this research lies in its meticulous data collection process. The inclusion of personal information along with detailed records of meal timings during both weekdays and weekends adds depth to the study, allowing for a comprehensive analysis of the subjects' chrononutrition patterns. Furthermore, the utilization of established assessment scales like the Zung Self-Rating Depression Scale and the Yale Food Addiction Scale enhances the validity and reliability of the findings.

The identification of a U-shaped association between eating mid-phase (EPFc), eating jetlag (EJL), and eating window (EW) with academic performance (GPA), depression (ZSDSI), and food addiction (FA) is a significant contribution to the field. This nuanced understanding of the relationship between chrononutrition and psychosocial well-being provides a valuable foundation for future research and interventions targeting adolescent health and academic success.

The study's conclusion, emphasizing the prevalence of circadian eating disorders in adolescents with poor academic performance, high levels of depression, and food addiction, highlights the practical implications of the findings. This insight could be instrumental in developing targeted interventions aimed at improving the overall well-being and academic outcomes of this demographic.

Overall, the article is a commendable piece of research that significantly advances our understanding of the intricate interplay between chrononutrition and adolescent well-being. The meticulous methodology, comprehensive data analysis, and practical implications make this article a valuable addition to the scientific literature. It is highly recommended for researchers, educators, and professionals in the fields of nutrition, psychology, and education. The authors are to be congratulated for their insightful contribution to this important area of study.

Reviewer 3 Report

Dear Autors,

You reached a great number of subjects and the purpose of your study may reach new insights for the research. But it takes too much energy to try understand you article because you mixture to much the concepts and in many situation you are not sufficiently clear to help understand the process. A great pedagogical effort is needed to help future lecturer easily reading you article. Most of our observation will have the objective to help you put this clearer and easy to read.

Our main concern: ZSDS and YFAS scales were translated and validated for Russian or local language? If yes specifies this and cite the articles of those translation/validation. If not, what was the procedure and witch guaranties it gives for the quality of the study?

Lines 36 to 38 you write: “Prolonged exposure to CS leads to an increased risk of developing chronic diseases (metabolic syndrome, obesity, type 2 diabetes, and cancer) and an accelerated aging process” Maybe it is better to write to CS misalignment.

Lines 67 to 88, It is not already sufficiently clear what is the lack of knowledge and what you propose to do to improve this situation. Please improve this part.

From material and methods to discussion, it takes too much energy to try understand what you did and what are the results. Because you mixture everything. Please in material and method do a pedagogical effort to let clear for future lecturer what you did, witch measure/instruments are used and how you calculate the results. Finish this part explaining the statistical procedures used. In this part do not present results, it has to be presented only in results parts, with all the figures and tables (not in material and not in discussion as you did). All the abbreviations used in the figure and table have to be presented extensively on the bottom of the table. The statistics used must be clearly specified in the tables, figure. The presentation of the results must be without discussion it is only writing what we can see after in the tables / figures. Using the table or figures and the presentation the future lecturers should find all the information’s necessaries to understand and interpret the results and their statistics. After in the discussion part you can discuss your results, this part in our opinion was easier to understand if you take out the tables, and do not need so big reformulation than in the other parts.

What is the Influence of physical exercise (sedentary vs sports peoples)? Not considering this in your questionnaire should be a limit for your study? Please discuss.

Why not analyse if they are differences between genders? What about the influence of latitude and the season? Please discuss a little bit.

After the conclusions, please write about: what are the practical implications of this study, what are the recommendation for future studies.

Wishes of success in the reformulation of your article!

Round 2

Reviewer 3 Report

Dear Authors,

thank you for your trust in the reformulation. 

For our main concern, we understand your justification and we think that future readers deserve to be informed. Then we suggest you to add the explanation you did for us: "In our study, the ZSDS and YFAS-C scales were used, translated into Russian. Comprehensive validation of Russian translations of the tests was not carried out. However, in previous studies, we have shown that (1) according to the Cronbach’s alpha tests have satisfactory internal consistency (this study as well as [1,2,4,5,6,7,8] from the list of references below); (2) there was a significant relationship between the number of symptoms of food addiction assessed using YFAS-C and the emotional type of eating behavior assessed using DEBQ [1,2], as well as a positive relationship between food addiction and the level of depression assessed using ZSDS [1,2,3,8]; and (3) the results of testing depression and food addiction using ZSDS and YFAS positively correlate with anthropometric indicators (BMI and WHtR) [1,2,3,5,7]. In general, all this indicates the fairly high reliability of these tools."
